# Surface Biofunctionalization of Tissue Engineered for the Development of Biological Heart Valves: A Review

**Wenpeng Yu** [1], **Ying Jiang** [1], **Feng Lin** [1], **Jichun Liu** [1,*] **and Jianliang Zhou** [2,*]

[1] Department of Cardiovascular Surgery, The Second Affiliated Hospital of Nanchang University, No. 1 Minde Road, Nanchang 330006, China
[2] Department of Cardiovascular Surgery, Zhongnan Hospital of Wuhan University, 169 Donghu Road, Wuhan 430071, China
[*] Correspondence: liujichun999@yeah.net (J.L.); zhoujianliang2010@163.com (J.Z.)

**Abstract:** Valve replacement is the mainstay of treatment for end-stage valvular heart disease, but varying degrees of defects exist in clinically applied valve implants. A mechanical heart valve requires long-term anti-coagulation, but the formation of blood clots is still inevitable. A biological heart valve eventually decays following calcification due to glutaraldehyde cross-linking toxicity and a lack of regenerative capacity. The goal of tissue-engineered heart valves is to replace normal heart valves and overcome the shortcomings of heart valve replacement commonly used in clinical practice. Surface biofunctionalization has been widely used in various fields of research to achieve functionalization and optimize mechanical properties. It has been applied to the study of tissue engineering in recent years. It is proposed to improve the shortcomings of the current commercial valve, but it still faces many challenges. This review aimed to summarize the modification strategies of biofunctionalization of biological heart valve surfaces based on tissue engineering to eliminate adverse reactions that occur clinically after implantation. Finally, we also proposed the current challenges and possible directions for future research.

**Keywords:** coating; surface biofunctionalization; tissue engineering; valve implants





## 1. Introduction

Valvular replacement is the most important treatment for patients with end-stage valvular disease [1]. More than 200,000 heart valve replacement surgeries are performed worldwide every year, and the number is expected to increase to 850,000 per year by 2050 [2]. During surgery, a natural valve that has lost its function is replaced with an artificial valve, either a mechanical heart valve (MHV) or a biological heart valve (BHV). MHV is generally made of pyrolyzed carbon, various polymers, and metal alloys, whereas BHV is generally composed of three types: (1) Chemically stable animal-derived tissue, (2) valve transplantation from cadaveric or living donors in the heart, or (3) patients' own valve transplantation from one location to another. Both MHV and BHV have their advantages and disadvantages [3–5]. Patients using mechanical valves require lifelong anticoagulation, which can lead to severe thromboembolism or bleeding complications. While the biorepairing of heart valves has achieved clinical success in the short to medium term, calcification leads to their eventual decay [6–9]. In addition, none of the heart valve replacements currently in use have shown the ability to grow or regenerate, which puts pediatric patients at risk of multiple surgeries. Therefore, creating a valve substitute with the ability to grow, repair, and regenerate using engineered tissue seems to be the best way to overcome many limitations at present.

The concept of tissue-engineered heart valves (TEHV) is described as the development of a heart valve with mechanical function and bioactivity under physiological conditions that facilitate the repair and reshaping of the scaffold [10]. TEHV generally addresses defects in implants by causing biomaterials to interact with autologous cells to achieve

growth and biointegration. An ideal heart valve prosthesis should be biocompatible and durable with antithrombotic and anticalcification effects and a physiologically hemodynamic profile [11].

This review was written to summarize the common modification strategies in developing biological valves based on engineered tissue, focusing on the development of non-glutaraldehyde cross-linkers, hydrogel coatings, and modification of nanocomposites. Finally, this review also proposed the current challenges and possible directions for future research.

## 2. Structure of Native Heart Valves

Structurally, the adult heart valve consists of three flexible tissue valves, approximately 23–26 mm in diameter and approximately 0.5 mm thick [7,12,13]. The leaflet structure is extremely complex, and the leaf cross-section is characterized by a three-layer structure, including ventricles, radially arranged collagen, and elastin fibrous tissue, each with its own extracellular matrix (ECM) protein composition and arrangement [14]. The fibrous tissue consists of a ring-like thick collagen bundle that provides mechanical strength to the leaflets [15]. Elastin fibrous stacks are composed of randomly oriented proteoglycans that act as a buffer between the fibrous muscle layer and the ventricular muscle layer [16]. The ventricles consist of radial elastin fibers (mainly elastin) with radial hyperelasticity, allowing the proper opening and closure of the valve [16]. Therefore, the heart valve leaflet has a multi-scale hierarchical organizational structure (from microscopic to macroscopic). Two main cell types are distributed in leaflets: Valve endothelial cells and valve stromal cells (VIC). Located at each layer of the ECM, VIC plays an important role in maintaining valve structure and function due to its ability to synthesize and remodel ECM [17–19]. In addition, the circumferential and radial arrangement of ECM proteins in the fibrous and ventricular layers imparts structural and mechanical properties to the anisotropy of the heart valve lobes [20]. These two important features of the heart valve lobe pose a major challenge to the use of traditional tissue-engineered stents in summarizing the structural characteristics of the valve and promoting the development of a new generation of bionic TEHV stents. The structure of a native heart valve is shown in Figure 1.

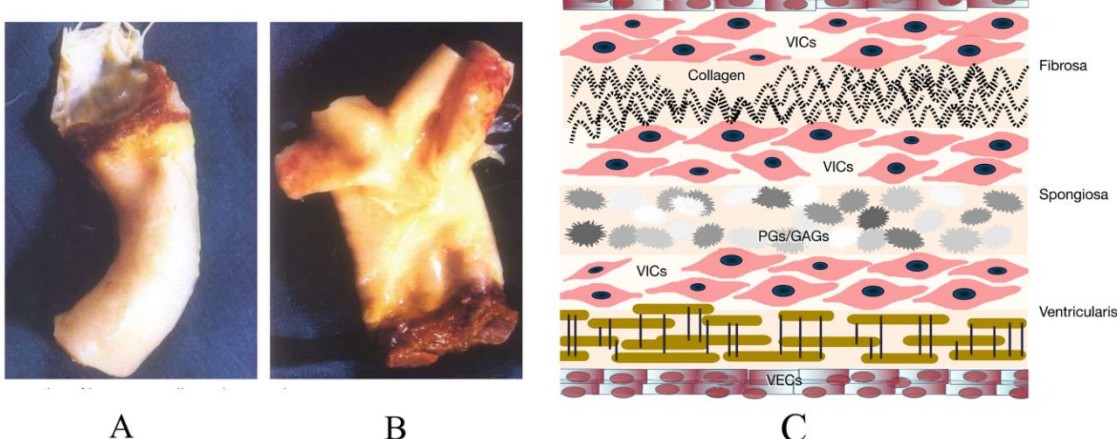

**Figure 1.** Photographs of human semilunar heart valves: Aortic (**A**) and pulmonary (**B**). (**C**) Detailed heart valve structure. It comprises three inner layers (ventricularis, spongiosa, and fibrosa) and an outer layer formed by valvular endothelial cells. The three inner layers contain mainly proteoglycans, glycosaminoglycans, collagen type I and type III, elastin, and valvular interstitial cells (adapted from Ref. [21]).

## 3. Surface Modification Methods

The advantages of surface modification compared with other technological methods lie in the simplicity and flexibility of the procedure, which does not change the properties of

the base material. Various physical and chemical modification techniques can be integrated to improve the overall effect [22,23]. The study of the materials' surface characteristics has also shown that the practical applicability of materials is, to some extent, influenced and limited by their surface features. Although the emergence of new materials is gradually filling gaps in certain areas as science and methodological design progress, it is impossible to synthesize new base materials for all applications [24]. It may therefore be quicker and more effective to modify some of the properties of existing materials to improve their performance or expand their applications, while increasing research into new materials. The usual methods of surface modification can be classified as physical and chemical methods. Physical material modification can be carried out by physical methods, such as physical adsorption or coating, to give the surface a certain roughness or pattern [25–27]. In addition, chemical surface modification generally utilizes a number of classical chemical reactions that result in binding to groups on the material surface. In contrast to physical methods, chemical methods use a chemical reaction in which the polymer is often bonded to the surface of the material in a chemical way. It is well known that chemical bonds are much stronger than intermolecular effects. As a result, the grafted layer can be more firmly bonded to the surface of the material by chemical reaction [28].

## 4. Biofunctionalization of the tissue Engineered Heart Valves

### 4.1. Non-Glutaraldehyde Cross-Linking

Glutaraldehyde (GLUT) is an aliphatic dialdehyde that can be attached to an amine functional group on collagen by the Schiff alkali reaction [29]; it has been shown to be reversible with easy hydrolysis [29,30]. GLUT or the presence of some groups such as aldehyde groups are highly cytotoxic and do not facilitate cell adhesion and growth on the valve, further leading to valve calcification [31,32]. In addition, GLUT can cause a strong immune response in the body, which leads to valve degeneration through the activation of macrophages, phagocytes, and T lymphocytes, and the aggregation of platelets [33–35]. Due to the defects in GLUT cross-linking, the implanted BHV is susceptible to structural damage and loss of function, resulting in a reduction in its durability and the need for reoperation in approximately only 15 years [36,37].

An ideal biomaterial cross-linker should be cytotoxic and less expensive. It can improve the mechanical properties of the material and inhibit calcification. Many fixed ECM-derived scaffolds of cross-linking agents are available, which can be divided into (1) chemical cross-linking agents and (2) natural cross-linking agents. Chemical cross-linkers include carbodiimide [1-ethyl-3-(3-dimethylaminopropyl)-carbondiimide (EDC)], epoxy compounds, six methylene diisocyanates, glycerin, and alginate [38–41]. Natural cross-linkers mainly include genipin, nordihydroguaiaretic acid, tannic acid, and proanthocyanidins [42]. Besides the aforementioned cross-linking agents, researchers have developed many uncommon cross-linking agents, and their effect is also worth looking forward to. Guo et al. [43,44] and Xu et al. [45] explored the comprehensive properties of the prepared BHV by radical polymerization and cross-linking, respectively. The results showed that the free radical polymerization cross-linking treatment had similar ECM stability and biaxial mechanical properties to the GLUT treatment. Furthermore, the samples showed better cytocompatibility, endothelialization potential, and anti-calcification potential and lower immune response in vivo after the free radical polymerization cross-linking treatment (Figure 2). Curcumin, quercetin, and other polyphenols have calcification inhibition potential similar to that of proanthocyanidins in cross-linked collagen and elastin scaffolds [46,47]. Curcumin [48] and quercetin [49] were developed as cross-linking reagents for valve materials. The cross-linked samples were superior to GLUT in terms of mechanical properties, blood compatibility, and cell compatibility.

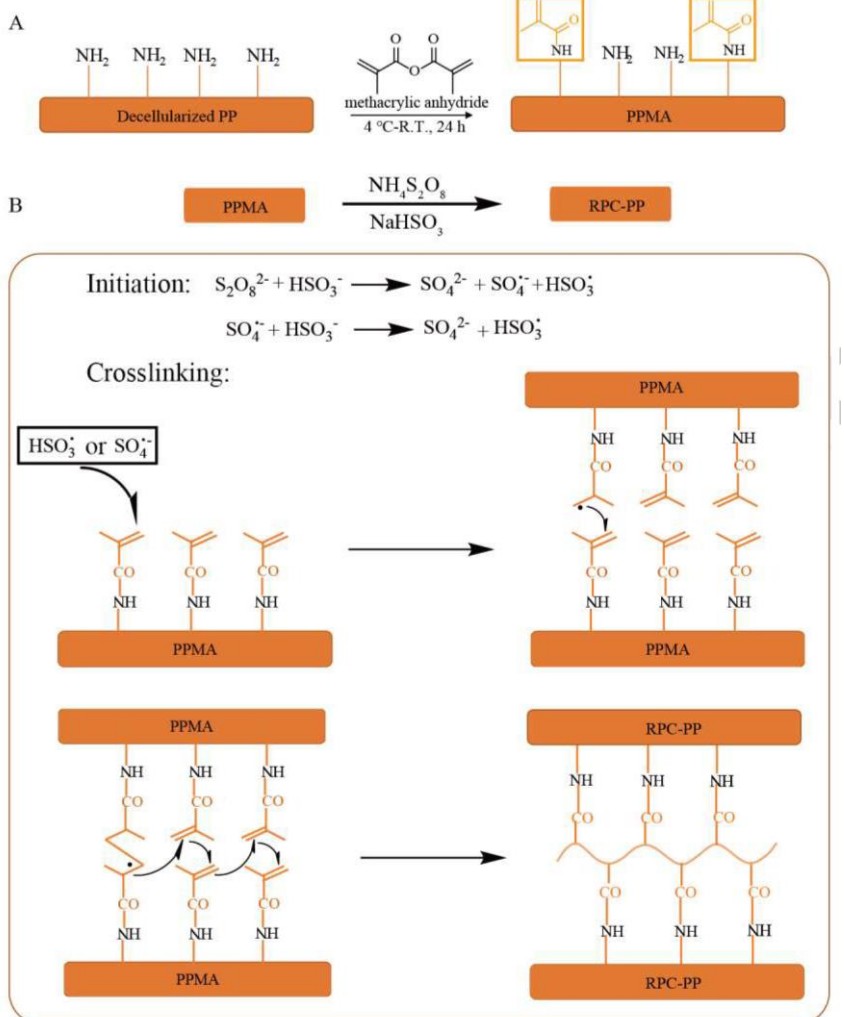

**Figure 2.** Schematic illustrations of the presumable cross-linking mechanism for RPC-PP. (**A**) Synthesis of PPMA. (**B**) Cross-linking of PPMA. PP, porcine pericardium; PPMA, porcine pericardium methacryloyl; RPC-PP, radical polymerization cross-linking-treated PP (adapted from Ref. [44]).

The focus of research on the bioprocessing of BHV involves the improvement of traditional GLUT cross-linking methods and the design of novel cross-linking methods. The special treatment employed by some researchers of BHVs cross-linked with ethanol, metal ions, -amino-oleic acid, and some other substances has effectively delayed the calcification of BHVs [50–54]. However, these methods still have many shortcomings. For example, epoxy cross-linked ECM-derived scaffolds are white, soft, and nonshrinking, with collagen remaining loose and natural [55]. Linear epoxy compounds exhibit low cross-linking and are also poorly stable due to their resistance to the degradability of collagen [55]. In addition, the epoxy compounds in the cross-linked extracellular matrix-derived scaffolds showed a degree of toxicity, leading to an immune response and eventual calcification failure [54,56,57]. The scaffold obtained by EDC cross-linking does not effectively inhibit calcification, which determines whether it can be used in the field of cardiovascular grafts [52]. The current substitution effect of cross-linkers has not been verified in the clinical stage. Therefore, although the development results of many cross-linkers are exciting, the ultimate success is yet to be achieved. Table 1 shows the basic research on non-glutaraldehyde cross-linking in the past 20 years.

**Table 1.** Application of different cross-linking agents in the study of tissue-engineered heart valves.

| Cross-Linking Agent | Advantages Compared with GLUT Treatment | Cross-Linking Mechanisms | Reference |
|---|---|---|---|
| Methacrylic anhydride | Improve the biocompatibility and anti-calcification performance | Radicals react with vinyl on collagen | [43–45] |
| Procyanidins | Anti-calcification | Hydrogen bond | [46,47] |
| Curcumin | Anti-calcification | Hydrogen bond | [48] |
| Quercetin | Increasing significantly the ultimate tensile strength and anti-calcification | Hydrogen bond | [49] |
| 1-Ethyl-3-(3-dimethylaminopropyl)carbodiimide | Improving the biocompatibility | Amide and ester bond formation of side groups | [58] |
| (3-glycidyloxypropyl) trimethoxysilane | Improving the cytocompatibility, endothelialization, hemocompatibility, and anti-calcification properties | Epoxy reacts with amino groups; inorganic polymerization | [59] |
| Dialdehyde pectin | Enhancing anti-calcification and anti- coagulation | Schiff base reaction | [60] |
| Alginate (oxidized alginate) | Improving the cytocompatibility, hemocompatibility, and anti-calcification properties | Amino reaction with carboxyl groups | [61] |
| Rose Bengal | Less cytotoxicity and better endothelialization potential | Photoinduced cross-linking | [62] |
| Triglycidyl amine | Anti-calcification | Reactive epoxy reacts with amine groups | [63–66] |

## *4.2. Modification Strategies after Decellularization*

The decellularized porcine aortic valve is a promising alternative to the ideal tissue-engineered valve scaffold [67–70]. Pretreatment of the porcine aortic valve with fixatives and detergents can greatly stabilize xenografts, improve their persistence, and reduce immunogenic updates. However, biofunctionalized decellularized porcine valves, such as Synerggraft, have failed due to decreased mechanical properties and poor cell aggregation [71]. Therefore, surface biofunction has attracted widespread attention as an important means to improve the performance of decellularized valves.

### 4.2.1. Hydrogel Coating

Physical coatings can give the substrate an ideal morphology and corresponding biological function, which is simple, fast, and efficient compared with chemical modification. In recent years, hydrogels have been widely used in engineered tissue due to their good biocompatibility and adjustable mechanical properties, but their application has been limited due to poor mechanical durability. Therefore, hydrogel coatings based on decellularized scaffolds have been developed by many researchers. These coatings provide good biocompatibility and function as carriers for decellularized scaffolds, while decellularized scaffolds provide sufficient mechanical properties. A variety of hydrogels, including natural (e.g., collagen [52], fibrin [72,73], and hyaluronic acid [73,74]), synthetic (e.g., polyethylene glycol [75,76] and polyvinyl alcohol [77]), and composite hydrogels (e.g., type I collagen with chondroitin sulfate [78]), have been developed, which provide good biocompatibility and bioactivity for engineered tissue applications. Synthetic hydrogels are composed of synthetic polymers, which have the advantages of adjustable mechanical properties and structure and easy control of chemical composition compared with natural hydrogels. Taking enhanced endothelialization as an example, the researchers seemed to prefer to load vascular endothelial growth factor (VEGF) into hydrogels as a core factor and coat decellularized scaffolds using natural hydrogels such as hyaluronic acid [79], elastin [80], and alginate [81]. The types and functions of synthetic hydrogels are much more complex. They not only focus on solving the side effects of heart valve implants but also make more de-

tailed adjustments to the mechanical properties and fatigue resistance of the hybrid scaffold. Luo et al. used methacrylic anhydride to modify the decellularized heart valves and applied a mixed hydrogel made of sulfa methacrylate and hyaluronic acid methacrylate to the surface to improve the anti-calcification performance of BHVs. This strategy could achieve both endothelialization and anti-calcification, thus helping improve the main drawbacks of the existing commercial BHV products [82]. In addition, Jahnavi et al. [83] developed a biological hybrid scaffold with a natural structure of non-cross-linked decellularized bovine pericardium coated with a layer of polycaprolactone-chitosan nanofibers, showing superior mechanical properties. These synthetic hydrogels solved individual problems to varying degrees, but for hydrogels, most of the studies did not take into account their own problems, that is, poor fatigue resistance, which leads to rapid damage during circulating hydrodynamic stress. Strengthening the mechanical properties of the overmolded hydrogel is a necessary condition for long-term protection. A dual-network hard hydrogel was developed in response to the anti-fatigue properties of single-mesh hydrogels. This gel penetrated and anchored the matrix of the decellularized porcine pericardium to form a strong and stable conformal coating, reduce immunogenicity, improve antithrombosis, and accelerate endothelialization (a schematic diagram is shown in Figure 3) [84]. In the last decade, researchers seem to have focused on the use of hydrogels in tissue-engineered heart valves, and as research continues to grow, various potential hydrogels have been proposed (Table 2). Hydrogel coatings are now gaining attention in this field, but most of these studies are currently in the initial research phase and have not reached the clinical stage. Hydrogel coatings still have a long way to go in this field, but the proposal of many research concepts has provided more ideas and perspectives for future researchers.

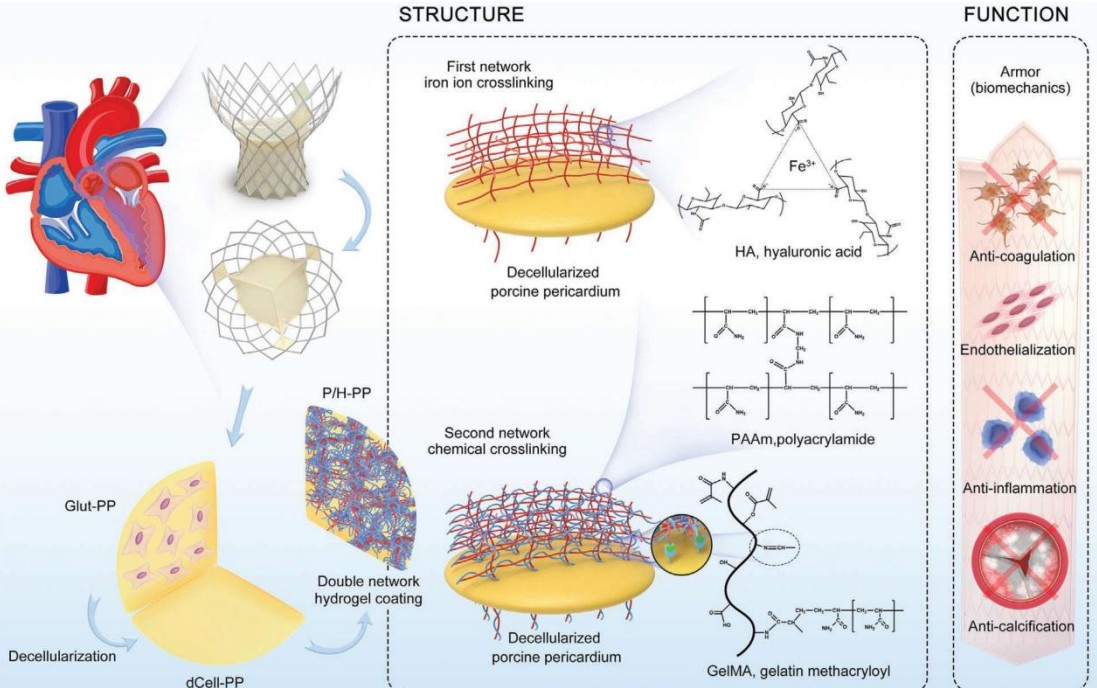

**Figure 3.** A schematic overview presenting the fabrication process, structure, and function of the decellularized porcine pericardium coated with polyacrylamide/hyaluronic acid dual-network hydrogel (adapted from Ref. [84]).

**Table 2.** Application of hydrogels in the study of tissue-engineered heart valves.

| Composition of Hydrogels | Advantages of Hydrogels | Mechanism | Reference |
|---|---|---|---|
| VEGF-Loaded hyaluronic acid hydrogel | Improve adhesion and growth potential, with less platelet adhesion and less calcification | Cross-linking the hydroxyl group of hyaluronic acid with the epoxide of BDDE | [79] |
| VEGF-loaded elastin hydrogel | Improve endothelialization potential | Cross-linking reaction between the amine groups of soluble elastin and hexamethylene diisocyanate | [80] |
| Sulfobetaine methacrylate and methacrylated hyaluronic acid | Improve endothelialization and anti-calcification properties | Radical polymerization | [82] |
| Hyaluronic acid and hydrophilic polyacrylamide | Improve endothelialization, biocompatibility, and anticalcififation properties | Ionic and chemical Cross-linking | [84] |
| SDF-1a-loaded MMP degradable hydrogel | Promote cell growth and mediate the tissue remodeling | Michael-type addition reaction | [85] |
| Polyhedral oligomeric silsesquioxane–polyethylene glycol hybrid hydrogel | Have anti-calcification potential | Formation of hydrogel network connecting POSS and MMP peptide using four-arm PEG-MAL | [86] |
| Chondroitin sulfate hydrogel | Improve endothelialization and shield against deterioration | Polymerization under UV lamps | [87] |

### 4.2.2. Cross-Linked with Nanocomposite

The advent of nanotechnology has made anti-tumor interventions possible. It also provides new ideas for TEHV. The nanoparticles have good biocompatibility. Furthermore, their surfaces have a variety of active groups, and researchers can modify them according to their purpose [88]. Common nanoparticles include lipids [89,90], polymers [91], inorganics [92,93], metals [94], and so forth. Nanoparticles are used in tissue-engineered heart valves mainly for gene transmission or the transmission of biologically active factors. This can effectively solve the limitation of cell seeding before TEHV implantation. A polyethylene glycol nanoparticle containing transforming growth factor-β1 was developed through a carbon diimide-modified fibrous valve scaffold that improved the ECM microenvironment of the tissue-engineered heart valves [95]. Based on this concept, we encapsulated VEGF in polycaprolactone (PCL) nanoparticles (Figure 4). Then, using the Michael addition reaction, PCL nanoparticles were introduced to the decellularized aortic valve, and a hybrid valve was prepared. The results showed that the mixed valve could effectively accelerate endothelialization [96]. A large number of nanoparticles used in tissue-engineered heart valves in recent years are biocompatible polymers, such as PEGs, PCL, and so forth. Recently, Hu et al. [97] were inspired by the natural biological systems and reported a new approach for cross-linking amino and carboxyl groups. The heart valves with erythrocyte membrane bionic drug-carrier nanoparticles were modified using amino and carboxyl groups remaining after GLUT cross-linking. This modified heart valve not only preserved the structural integrity, stability, and mechanical properties of GLUT-treated BHV but also significantly improved anticoagulation, anti-inflammatory, anti-calcification, and endothelialization properties.

An inorganic nanoparticle, mesoporous silica, has attracted much attention because of its advantages of good compatibility and adjustable particle size, pore size, and structure [98,99]. Pinese et al. [100] applied MSNs to tissue engineering and prepared siRNA/MSNPEI complexes for siRNA delivery. The results showed that siRNA/MSNPEI complexes were more efficient and less toxic than the traditional silencing methods. However, MSN has not been used in decellularized tissue engineering so far. Hence, this may be one of the key directions to be explored by researchers. Many metal nanoparticles, such as gold, silver, and titanium nanoparticles, have been widely examined in other fields. Their

future performance in TEHV based on their basic properties is also worth examining. These nanoparticles can be used as a functional platform to achieve the transmission of a certain substance (Table 3 lists the basic research of nanocomplexes applied to tissue-engineered heart valves in the last 15 years). However, they seem to be more flexible in terms of modification and function and can even be used as a cross-linking agent to make the structure more stable.

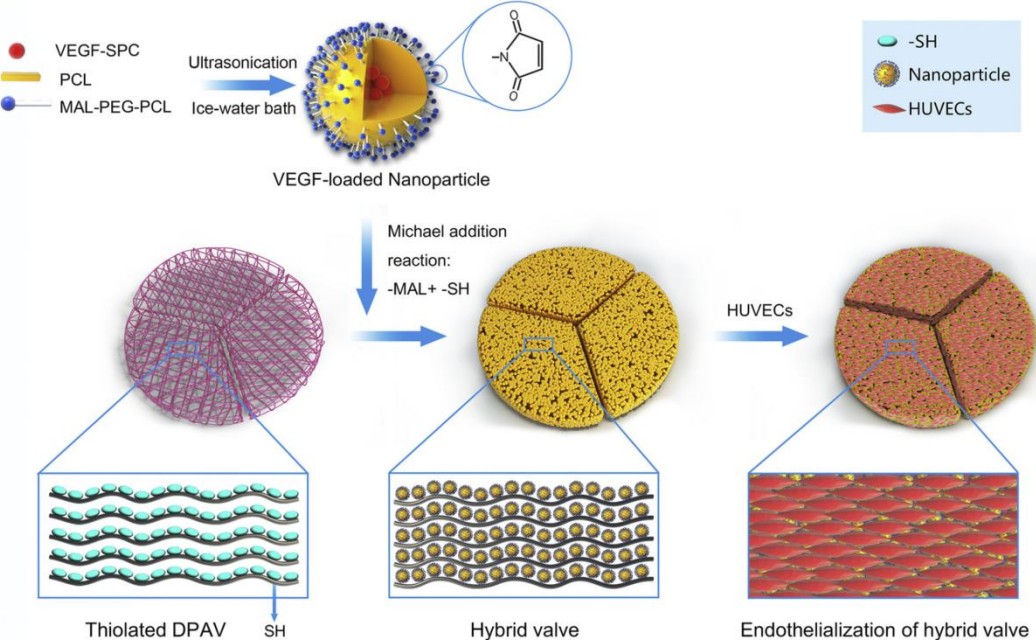

**Figure 4.** Preparation scheme of VEGF-loaded nanoparticles and endothelialization of the hybrid valve (adapted from Ref. [96]).

**Table 3.** Application of different nanocomposites in the study of tissue-engineered heart valves.

| Composition of Nanocomposite | Advantages of Nanocomposite | Mechanism | Reference |
|---|---|---|---|
| TGF-β1-loaded polyethylene glycol nanoparticles | Advantageous biocompatibility | Combining with PEG nanoparticles by carbodiimide | [95] |
| VEGF-loaded polycaprolactone nanoparticles | Acceleration of endothelialization | Michael addition reaction | [96] |
| RBC-based rapamycin and atorvastatin calcium-loaded poly(lactic-*co*-glycolic) acid nanoparticles | Anticoagulation, anti-inflammation, anti-calcification, and endothelialization properties | Amidation reaction | [97] |
| OPG-loaded polycaprolactone nanoparticles | Anti-calcification | Michael addition reaction | [101] |
| Rivaroxaban-loaded nanogels | Acceleration of endothelialization and antithrombogenicity | Self-assembly | [102] |
| Polyhedral oligomeric silsesquioxane-nanocomposite | Acceleration of endothelialization potential | Reaction of the silanol groups of cyclohexanechlorohydrine-functionalized POSS with isocyanate | [103–105] |

## 5. Summary and Future Perspectives

Recent advances in surface biofunctionalization enable the functionalization of TEHVs to improve their performance in biomedical applications at the interface of biological macromolecules, cells, tissues, and biomaterials. This review summarized the common modification strategies in developing biological heart valves based on tissue engineering,

focusing on the development of non-glutaraldehyde cross-linkers, hydrogel coatings, and the modification of nanocomposites. Of the three research directions mentioned earlier, non-glutaraldehyde cross-linkers were first explored. However, the modification of coatings and nanocomposites is the most complex and flexible method for TEHV.

Tissue engineering can make substantial progress with a certain degree of biological function to accommodate the unique challenges inherent in valve design. Notably, this requirement is the most acute among children, whose options are currently quite limited. Moreover, although the TEHV studies cited in this review provided some understanding of surface biofunctionalization during the development of TEHV, our efforts to date have been largely empirical, and hence the mechanisms influencing the maturation process remain unknown. The challenges faced by researchers involved in engineering heart valves are significant and comprehensively address many areas of expertise (Figure 5).

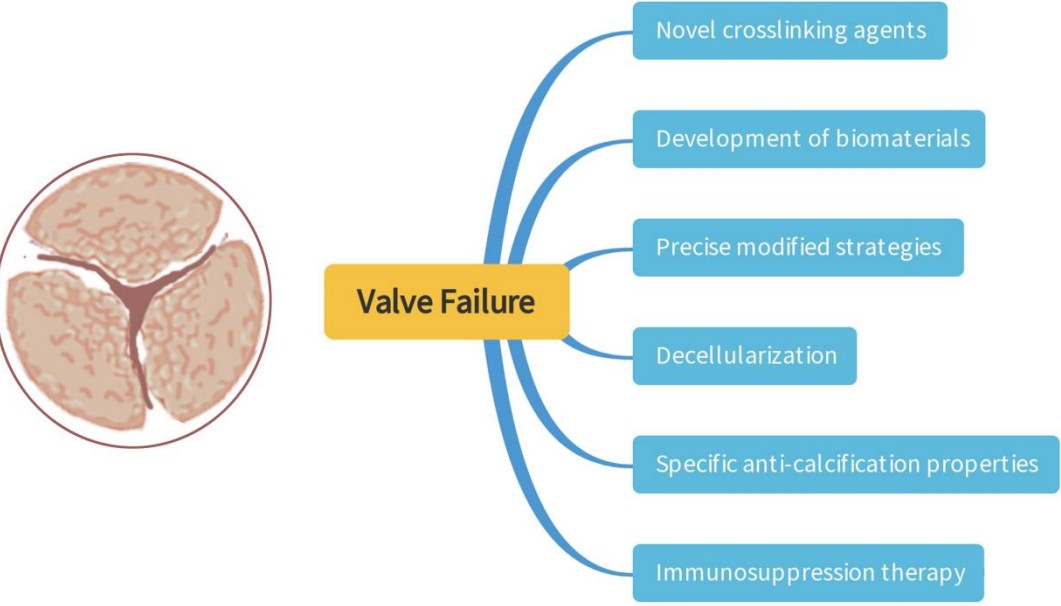

**Figure 5.** Key factors of TEHV development and strategies to retard TEHV.

The chemical surface modification of polymer biomaterials has received widespread attention, due to the global emphasis on medical implants and devices, and hence its future seems promising. Although previous research has primarily stayed in the laboratory stage, technology can be extended to the industrial level by employing strategies to modify polymer biomaterials, thus opening the door to a wider range of biomedical applications. We have reason to believe that the successful pursuit of various biomedical applications can eventually be achieved with the continuous development of chemical surface modification technology for polymer biomaterials.

**Author Contributions:** Methodology, J.Z. and J.L.; writing—original draft preparation, W.Y.; writing—review and editing, Y.J. and F.L. All authors have read and agreed to the published version of the manuscript.

**Funding:** This work was funded by the Natural Science Foundation of Jiangxi Province (grant No. 20204BCJ22028), the National Natural Science Foundation of China (grant No. 81860079, 81770388), and the Natural Science Foundation of Jiangxi, China (No. 20192BAB205004, No. 20192ACBL20036).

**Institutional Review Board Statement:** Not applicable.

**Informed Consent Statement:** Not applicable.

**Data Availability Statement:** Not applicable.

**Conflicts of Interest:** The authors declare no conflict of interest.

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
