# Peer review of "Surface Biofunctionalization of Tissue Engineered for the Development of Biological Heart Valves: A Review"

_coatings, doi:10.3390/coatings12091322_

Round 1

Reviewer 1 Report

The manuscript is features an interesting retrospect on how far heart valve tissue engineering has come in the past decades. However, sentence construction and composition still requires improvement to make it more readable. Grammar checking by either a native English speaker or through an English language center is highly recommended. It should be noted that “tissue engineering heart valves” and “tissue engineered heart valves” mean differently. The former is an active verb form while the latter is an object form.

Title of the article is recommended to be modified to: “A Review of Surface Biofunctionalization for the Development of Tissue Engineered Heart Valves.”

The authors should state how the articles included in the reference were selected. What year range of publications are included (5 years/10 years/20 years)? Are these selected based as basic/ advanced research or does it include clinical data? Limitations with regards of the scope should be clearly stated as to clearly delineate the topics included in the article.

The current outline of the manuscript is not effective. While sections 1-3 is a good start, the authors failed to properly ease the transition of the discussion towards section 4 and beyond.  Some re-structuring is required to better present the body of the review in relation to the existing studies. It is quite unclear whether the authors are primarily discussing modification of decellularized tissues or includes synthetic based tissue engineered implants because both sections 4.2.1 and 4.2.2 are solely discussed under Modification strategies after decellularization. This could be resolved by properly stating the scope and limitation of the review.

Reviewer 2 Report

The manuscript entitled 'Surface biofunctionalization of tissue engineering for the development of biological heart valves: A review' is a well structured review discussing the structure of native heart valves, surface modification strategies, biofunctionlization of tissue engineered heart valve and its future prospect. Figures and tables are adequate, however, it is suggested to add more recent findings in the tables.

Reviewer 3 Report

The review was written well and the information is important for the people in the related research fields. However, there are few minor correction can be done to improve the quality of the review. Please refer to the annotated PDF file.

Round 2

Reviewer 1 Report

The authors have satisfactorily accomplished the suggested changes. No further modifications are recommended.